# Verified Path Following Using Neural Control Lyapunov Functions

**Alec Reed**     **Guillaume Berger**     **Sriram Sankaranarayanan**     **Christoffer Heckman**

Department of Computer Science
University of Colorado Boulder, USA.
`first.lastname@colorado.edu`

**Abstract:** We present a framework that uses control Lyapunov functions (CLFs) to implement provably stable path-following controllers for autonomous mobile platforms. Our approach is based on learning a guaranteed CLF for path following by using recent approaches — combining machine learning with automated theorem proving — to train a neural network feedback law along with a CLF that guarantees stabilization for driving along low-curvature reference paths. We discuss how key properties of the CLF can be exploited to extend the range of the curvatures for which the stability guarantees remain valid. We then demonstrate that our approach yields a controller that obeys theoretical guarantees in simulation, but also performs well in practice. We show our method is both a verified method of control and better than a common MPC implementation in computation time. Additionally, we implement the controller on-board on a $\frac{1}{8}$-scale autonomous vehicle testing platform and present results for various robust path following scenarios.

**Keywords:** Path Following, Trajectory Tracking, Control Lyapunov Functions, Plan Execution, Verified Autonomy.

## 1   Introduction

Path following is a fundamental primitive for autonomous robotic platforms, wherein the goal is to steer the platform along a desired path in the workspace. This is important for mobile platforms since many approaches to platform design rely on the separation between high-level planners that search over paths without considering platform dynamics and lower-level plan execution which implements path following [1, 2]. However, path following is a challenging problem since platform models of interest are nonlinear. Popular approaches such as the "Stanley" controller (originally proposed by Hoffman et al. for the 2005 DARPA grand challenge [3]) can become unstable depending on the choice of the controller's parameters and the curvature of the path [4]. Similarly, feedback controllers based on linearization can also be unstable if the deviation is too large [5].

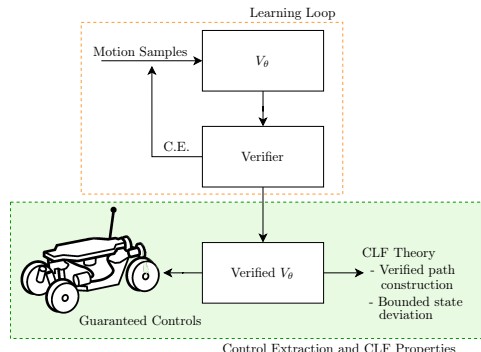

Figure 1: Learning CLFs. Motion samples consist of state-control pairs used to train the neural network $V_\theta$ with parameter $\theta$. The Verifier produces counterexamples (C.E.) until $V_\theta$ satisfies the specifications. The green box contains our contributions: 1) extraction of control inputs from $V_\theta$ that ensure bounded path deviation, 2) constructing advanced maneuvers using a single CLF, and 3) demonstration on a real platform.

Control Lyapunov functions (CLFs) are a general approach for controller design, typically for steering a control system towards an equilibrium point [6, 7]. A CLF is a positive definite function such that for any state within a well-defined basin of attraction, there exists a control input that causes its derivative to be negative. Thus, CLFs naturally define stabilizing controllers [8, 9]. However, the generation of CLFs has remained a challenging problem for researchers, and various CLF gen-

6th Conference on Robot Learning (CoRL 2022), Auckland, New Zealand.

eration methods have been developed. Neural networks have recently become a popular method for designing CLFs given their ability to converge even on systems with sophisticated dynamics. The idea is to use machine learning algorithms to infer candidate CLFs by minimizing a special loss function called "Lyapunov risk" in combination with automated theorem-proving algorithms to verify the results of the learning algorithm. As a result, if successful, the approach yields a CLF as well as a stabilizing controller. Thus, neural CLF generation is a promising approach for designing controllers for nonlinear systems. Nevertheless, its use has been limited to stabilization to fixed equilibria, whereas our application seeks to stabilize along a path. Secondly, empirical validation on an actual robotic test platform has not been publicly performed.

In this paper, we address the path-following problem for a path in the workspace whose curvature $\kappa(s)$ varies with the distance $s$ along the path. For $\kappa(s) \equiv 0$, which corresponds to a straight-line path, the problem is amenable to existing neural CLF synthesis tools, but it becomes progressively harder as the path curvature increases and becomes dependent on $s$.

To address this, we exploit key properties of the system and the generated CLF to design a formalism wherein the controller derived for a straight-line path can be used for paths with non-zero, varying curvature. We derive bounds on the maximum path curvature for which the controller is valid and we show that we can construct complex maneuvers using this approach.

Next, we demonstrate our approach on a $\frac{1}{8}$-scale vehicle platform. Our demonstrations show that (a) the model-based design approach transfers to a real robotic platform even though the model used for learning the CLF is a simpler model; and (b) the resulting approach is favorable compared to other verified and unverified path-following methods in both curved and straight path-following scenarios.

## 1.1 Related Work

Numerous methods have been studied for generating CLFs. Certain nonlinear control design methods such as backstepping automatically synthesize CLFs but are not widely applicable. Tan and Packard studied an approach using bilinear constraints that are hard to solve in practice [10]. Majumdar et al. propose a similar approach for path following but use a control feedback law derived through linearization as a heuristic initialization to solve bilinear constraints efficiently in practice [11]. Data-driven approaches for CLFs were first studied by Khansarizadeh et al. [12]. However, a lot of data is potentially needed to generate CLFs and the final result needs to be verified against the system dynamics. Previous work by Ravanbakhsh et al. [13] showed an active approach for deriving CLFs for nonlinear systems that was adapted to path following using the idea of control funnel functions [14, 15]. Furthermore, their approach was shown to be effective on a autonomous vehicle testbed in the lab. However, the techniques described so far focus mostly on deriving polynomial CLFs. It is well known that CLFs can be non-polynomial even for polynomial dynamical systems. Recently, neural networks have been used to generate CLFs for systems with complex, non-polynomial dynamics. Introduced by Richards et al. [16] and Chang et al. [17], neural networks have been shown to be an effective method for generating CLFs for a variety of systems.

Majumdar et al. [18] provide the framework of "funnel libraries" in which a set of funnels $\mathcal{F}$ and associated feedback controllers are generated. This library of funnels can be sequenced together to form complex vehicle maneuvers. Bouyer et al. [19] showed that this approach can reduce the reach-avoid problem to that of a timed automata by switching between appropriate controllers based on state measurements. Funnel libraries are an interesting approach towards a more rigorous plan execution that supports a motion planner which can restrict itself to maneuvers available in the library. However, maintaining such a library can be cumbersome. In particular, a large number of controllers may be needed to cover the space of possible maneuvers. The approach presented in this paper can simplify funnel libraries by covering a larger set of trajectories with fewer controllers.

## 2 Preliminaries: Problem Definition and CLF Learning

Consider the problem of stabilizing a dynamical system $\dot{x} = f(x, u)$, wherein $x \in \mathcal{D} \subseteq \mathbb{R}^n$ and $u \in \mathcal{U} \subseteq \mathbb{R}^m$, to the equilibrium point $x = 0$ without loss of generality. This is a typical control problem, e.g., where a mobile platform has command inputs that can lead it to follow a specific trajectory which may be parametrized by a point in time. Let $V : \mathcal{D} \to \mathbb{R}$ be a control Lyapunov function (CLF) for $f$ on $\mathcal{D}$. Formally, a CLF is defined in Definition 1.

**Definition 1.** *A CLF is a continuously differentiable function $V : \mathcal{D} \to \mathbb{R}$ that is positive definite, i.e., $V(0) = 0$ and $V(x) > 0$ for all $x \neq 0$, and satisfies the following decrease condition*

$$\forall\, x \in \mathcal{D} \setminus \{0\}, \; \exists\, u \in \mathcal{U} : \; \nabla V(x) \cdot f(x, u) < 0. \tag{1}$$

Rather than requiring that the system reaches an equilibrium, we are often interested in region stability, wherein we define a neighborhood $\mathcal{I}$ of 0, and require that the Lyapunov condition Eq. (1) is satisfied in the region $\mathcal{D} \setminus \mathcal{I}$, called the *Lyapunov (satisfiability) region*. Using the CLF $V$, control inputs can be generated for any state $x$ that belongs to the largest level set of $V$ contained completely within $\mathcal{D}$ to drive the system to the smallest level set of V that contains $\mathcal{I}$, in finite time [19].

The synthesis of CLFs is known to be very challenging, and the complexity increases rapidly with that of the system dynamics and the control task. The core idea of this work is to synthesize CLFs for simple maneuvres for the robot platform (namely, straight-path following) and then extend these to more complex maneuvres. To synthesize the "simple-task" CLFs, we used the approach discussed in Chang et al. [17], which is a neural network based learning solution for generating CLFs. In this framework potential neural CLFs are trained using state-control samples from the defined Lyapunov region $\mathcal{D} \setminus \mathcal{I}$ and then verified by an SMT solver to ensure that the candidate CLF meets the conditions in Definition 1. Details of the CLF generation process are provided in appendix A.

**Region of Attraction**  While the target of CLF training is to generate a CLF that is valid inside the Lyapunov region, an important metric is the region of attraction (ROA) [20].

**Definition 2.** *Let $V$ be a CLF with domain $\mathcal{D}$ for a system $f(x, u)$. Any level set of $V$ completely contained in $\mathcal{D}$ defines a region of attraction. That is, for any $\beta > 0$, if $\{V(x) \leq \beta\} \subseteq \mathcal{D}$, then $\{V(x) \leq \beta\}$ is an ROA for the system. The maximal region of attraction is the maximum level set of $V$ contained within $\mathcal{D}$.*

Note that it is important to consider the largest level set contained in $\mathcal{D}$. Lyapunov conditions do not directly ensure safety (i.e., infinite-time horizon containment) for any states in the verified Lyapunov region since large level sets of $V$ may exit $\mathcal{D}$ while descending the gradient. Upon exiting $\mathcal{D}$ there is no guarantee that actions will be available to reduce $V$. To ensure safety, a system's initial state must be contained within the maximal region of attraction.

### 2.0.1   Vehicle Model

To train the system we use a path-following model with a fixed velocity. The model consists of two states: $d_e$ and $\theta_e$, where $d_e$ is the distance from the line as measured from the rear axle and $\theta_e$ is the theta offset from nearest point on the target line (see Figure 2). The dynamics of this system are:

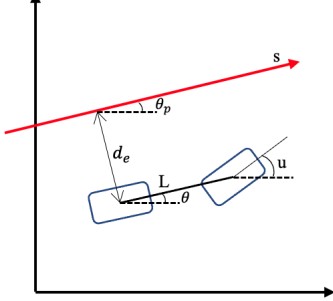

$$\dot{d}_e = v \sin(\theta_e)$$
$$\dot{\theta}_e = \frac{v \, \tan(u)}{L} - \frac{v \, \kappa(s) \, \cos(\theta_e)}{1 - d_e \kappa(s)}, \tag{2}$$

where $\theta_e = \theta - \theta_p$ and $\kappa(s)$ is the curvature of the reference path at the nearest point. Note that $\kappa(s)$ is 0 for the straight-line path shown in Figure 2.

Figure 2: Diagram of the states of the wheeled vehicle described in Equation 2, $s$ being the parameterized curve.

To learn a neural CLF using the approach in Appendix A, state-control samples are generated for the learning algorithm. To calculate $u$ at any given sample $(d_e, \theta_e)$, we use an LQR controller.

### 2.1   Straight Path to Curved Path Following

Finding CLFs for the vehicle model described in Eq. (2) for paths of various curvature would (a) be expensive and potentially infeasible (as the neural CLF training algorithm offers no guarantee of finding a CLF and the problem becomes more difficult as the curvature increases); and (b) lead to only a finite set of curves that the vehicle can execute. Instead, our approach consists of finding

a single CLF for the straight-line path, and compute the maximum curvature for which the CLF is valid, in order to extend the CLF to paths with nonzero, possibly varying, curvature.

If the minimal required steering input for the straight-path CLF is smaller than the maximum allowed steering input $u_{\max}$, we can derive a curved-path CLF. More precisely, given a CLF $V$ for a straight-line path, i.e., for $\kappa(s) = 0$ in Eq. (2), with ROA contained in the region $d_e \in [-d_{\max}, d_{\max}]$, we define the minimal required input $u_{\text{req}}$ for $V$ as the smallest value of $u$ satisfying that for all state $x$ in the Lyapunov region, there exists a steering input $u$ with $|u| \leq u_{\text{req}}$ that decreases the value of $V$, i.e., $\nabla V(x) \cdot f(x, u) < 0$. If $u_{\text{req}} < u_{\max}$, then one can exploit the gap $\tan(u_{\max}) - \tan(u_{\text{req}})$ to follow a path with curvature $\kappa(s)$ bounded as follows.

**Proposition 1.** *If* $|\kappa(s)| \leq \frac{\tan(u_{\max}) - \tan(u_{\text{req}})}{L + d_{\max}}$, *then* $V$ *is a CLF for Eq. (2).*

*Proof.* Note that for all $d_e \in [-d_{\max}, d_{\max}]$, $\left|\frac{\kappa(s)\cos(\theta_e)}{1 - d_e\kappa(s)}\right| < \frac{|\kappa(s)|}{1 - d_{\max}|\kappa(s)|}$. Thus, if $\frac{|\kappa(s)|}{1 - d_{\max}|\kappa(s)|} < \frac{\tan(u_{\max}) - \tan(u_{\text{req}})}{L}$, one can use the steering input $u(s) = \arctan(\tan(u) + \frac{L|\kappa(s)|}{1 - d_{\max}|\kappa(s)|})$, where $u \in [-u_{\text{req}}, u_{\text{req}}]$ satisfies $\nabla V(x) \cdot f(x, u) < 0$. It holds that $|u(s)| \leq u_{\max}$. Moreover, with this input the second equation of Eq. (2) becomes $\dot{\theta} = \frac{v \tan(u)}{L}$. Since $u$ is a decreasing input for $V$ with $\kappa(s) = 0$, it is clear that $V$ is a CLF for curved-path following with input $u(s)$. $\square$

Similar to a library of funnels [18], the CLF generated for straight-path following can be used for any path with curvature $|\kappa(s)| \leq \frac{\tan(u_{\max}) - \tan(u_{\text{req}})}{L + d_{\max}}$ per Proposition 1. However, while previous works require switching controllers [19, 18], our solution relies on a single CLF for the duration of the maneuver.

## 2.2 Control Extraction

---
**Algorithm 1** Vehicle Rollout CLF Minimization
---
import CLF, set vehicle model parameters, generate target path, discretize steering action space, set $dt$ (time step), set $K_v$ (control cost)
**while** sim == true **do**
    **for** steer in steeringAngles **do**
        $\theta_e, d_e \leftarrow calcError(updateState(currentState, steer))$
        $vxRollouts.append(calcVx(\theta_e, d_e))$
    **end for**
    $u \leftarrow steeringAngles[argmin(square(steeringAngles) * K_v + vxRollouts)]$
**end while**

---

The definition of a CLF (Definition 1) guarantees the existence, at any state in the Lyapunov domain, of a control input causing the CLF to decrease over infinitesimal time horizon (negative Lie derivative). A simple control strategy is then to simply execute the control that minimizes the Lie derivative at the current state. One can plot a theoretically executable path to descend the CLF as quickly as possible using this method. However this approach usually results in saturation of the control surface. In our system, this holds true as the Lie derivative is a monotonic function of $u$, so that the control that minimizes the Lie derivative of the CLF is always the maximum or minimum $u \in \mathcal{U}$. Such a bang–bang controller [21] is not desirable in practice because abrupt changes in the input can damage the platform, and requires a high actuation frequency.

A better practical method for control extraction is to simulate some actions and select the action that causes the largest reduction of $V(x)$. Given Eq. (1) and provided the time horizon is small enough, there always exists a control action to reduce the CLF if the vehicle is in the Lyapunov region of the CLF. The addition of a small control cost can be included for smoother operation. This approach allows for stabilization to the origin, as well as more realistic models to be introduced to compute realistic maneuvers. A simple, functional implementation of this idea is outlined in Algorithm 1. This algorithm can be expanded to operate like a traditional MPC [22] by increasing the look-ahead, and minimizing a cost function balancing future $V(x)$ value, control cost, and terminal $V(x)$ value.

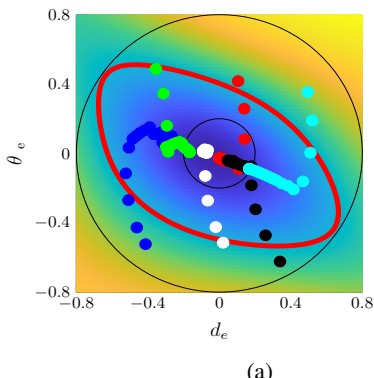
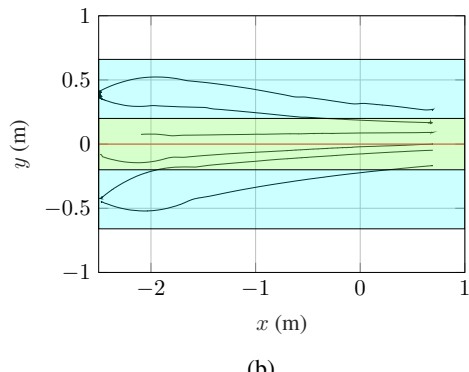

(a)                                                    (b)

Figure 4: Results of CLF minimization on the testing platform. (a) Colored points are sampled platform trajectories given various starting positions and orientations, black lines define the Lyapunov region and the red line the maximal ROA. Surface colors show the value of $V(x)$, orange being larger, blue smaller. (b) Vehicle paths at starting states $(d_e, \theta_e)$: $(0.4, 0.4)$, $(0.4, -0.4)$, $(0.0, 0.4)$, $(0.0, -0.4)$, $(-0.4, 0.4)$, $(-0.4, -0.4)$. The cyan area contains the ROA in the direction of $d_e$, the green area is the lower bound of the Lyapunov region of $d_e$ and the red line is the target path and black lines are platform paths.

# 3 Implementation and Experimental Evaluation

## 3.1 CLF Generation

Our neural Lyapunov control framework was successful in generating CLFs for various target paths and vehicle parameters. Figure 4(a) shows a generated CLF for a straight-line path. The CLF is verified inside the Lyapunov region defined by an inner and outer radius of 0.2 and 0.8 respectively. Note that while the CLF is valid in those bounds, the maximal ROA is the only region in which we can guarantee safety, although in practice we observe success, in both converging to the minimum and remaining in the Lyapunov region, outside the ROA.

## 3.2 Controller Implementation and Evaluation

**Platform**     To test our approach in the real world we use a $\frac{1}{8}$-scale vehicle platform in a motion capture space. The wheel base of the vehicle is 34 cm and the velocity for all tests is fixed at 0.5 m/s. The vehicle includes an on-board computer for real-time computation and receives pose estimates from the motion capture system through a WiFi connection. After computation, the control action is transmitted to a control unit which controls the steering servos and motor of the vehicle. Remarkably, each iteration of Algorithm 1 is executed in less than 150 $\mu$s.

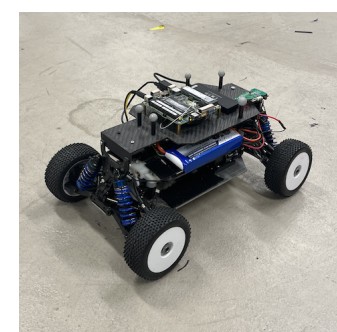

Figure 3: Vehicle Testing Platform

**Control Extraction in Real Time**     The test objective is to use the generated CLF to stabilize the platform to a straight-line path from $x = -2.5$ m to $x = 1$ m at a constant velocity of 0.5 m/s. On the real-world platform the Lie derivative minimization control extraction method is unable to stabilize the platform to a straight-line path: since the control surface is always saturated, the platform continuously overshoots the distance to the target path, and therefore is unable to stabilize.

Using Algorithm 1 we simulate potential rollouts for the platform and execute the lowest-cost action. We discretize the continuous action space $[-\frac{\pi}{4}, \frac{\pi}{4}]$ to 31 equally-spaced steering values. We use the kinematic bicycle model to project our current state a single time-step into the future, setting the time-step to be equal to our system average update rate $dt = 150$ $\mu$s.

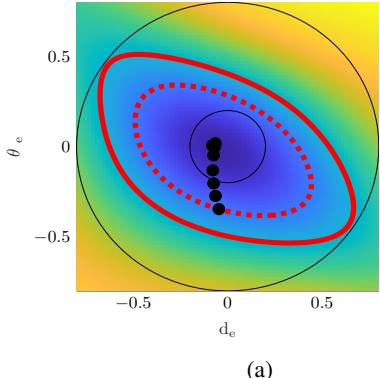

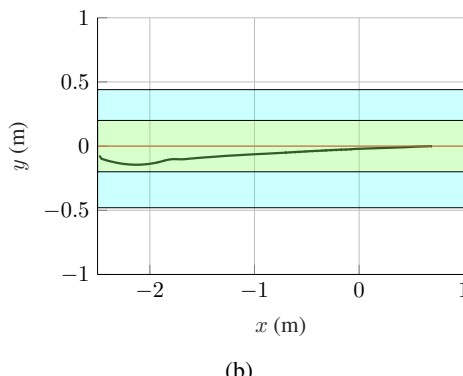

|     |     |
| --- | --- |
| (a) | (b) |

Figure 5: (a) Reduced invariant region based on starting point $(0, -0.4)$ where the solid red line is the maximal ROA and the dashed red line is the reduced invariant region at $(0, -0.4)$. (b) Computed $d_e$ boundary when platform starts at $d_e = 0$, $\theta_e = -0.4$.

Figure 4(a) shows the convergence to the minimum of the CLF using Algorithm 1, starting from various positions in the Lyapunov region. In practice, while not guaranteed to converge, we see that even positions starting outside the maximal ROA converge to $V(x) = 0$.

Given any starting point in the maximal ROA we can plot the guaranteed maximum $d_e$ and $\theta_e$ by computing the level-set of $V(x_{start})$ and taking the $\max \theta_e$ and $\max d_e$ of the computed set. Figure 5(a) shows a reduced invariant region based on $x_0 = (0, -0.4)$. For this particular starting position we can guarantee the platform will deviate no further than $[-48, 44]$ cm from the target path (a 27% $d_e$ reduction when compared to the maximal ROA maximum $d_e$ of 0.66).

**Maximum Curvature Computation**  In Subsection 2.1, we showed that a CLF for straight-path following can be extended to curved-path following if the range of input required for straight-path following is smaller than the maximum available steering input $u_{\max} = \frac{\pi}{4}$. The maximum allowed curvature depends on gap $= \tan(u_{\max}) - \tan(u_{\mathrm{req}})$, where $u_{\mathrm{req}}$ is the smallest range required for straight-path following. Given the CLF $V$ computed Subsection 3.1, $u_{\mathrm{req}}$ can be computed as:

$$u_{\mathrm{req}} = \max_{d_e, \theta_e} \min \left\{ |u| : \frac{\partial V}{\partial d_e}(d_e, \theta_e) v \sin(\theta_e) + \frac{\partial V}{\partial \theta_e}(d_e, \theta_e) \frac{v \tan(u)}{L} < 0 \right\}, \qquad (3)$$

where the maximum is taken for $d_e, \theta_e$ in the Lyapunov region. Eq. (3) can be rewritten as a single maximization problem:

$$\frac{1}{L} \tan(u_{\mathrm{req}}) = \max_{d_e, \theta_e} \left| \frac{\frac{\partial V}{\partial d_e}(d_e, \theta_e) \sin(\theta_e)}{\frac{\partial V}{\partial \theta_e}(d_e, \theta_e)} \right|. \qquad (4)$$

Eq. (4) has only two variables $(d_e, \theta_e)$, so that it can be solved easily using for instance grid search, or nonlinear solvers like Ipopt [23]. We compute the value $\frac{1}{L} \tan(u_{\mathrm{req}}) = 0.496$. With $L = 0.3$, we get that gap $= 0.8512$. By Proposition 1, with $d_{\max} = 0.8$, it follows that $V$ can be extended to curved-path following for any path with curvature $\kappa(s) \leq \frac{\mathrm{gap}}{L + d_{\max}} = 0.774$, i.e., with radius of curvature $R \geq 1.3$ m.

Using the approach discussed in Section 2.1 we can execute a complex object avoidance maneuver. The maneuver is a curved path with varying curvature, bounded by $0.4$ m$^{-1}$ (the minimal radius of curvature is 2.5 m); see Figure 6. The figure shows platform paths of various starting positions and orientations after executing Algorithm 1 on the constructed path.

### 3.3  Side-by-side comparison with alternative techniques

We compare our control method against two MPC implementations [22] (a non-verified controller) as well as the "deep learned" controller trained when computing the neural CLF (DLC) (a verified

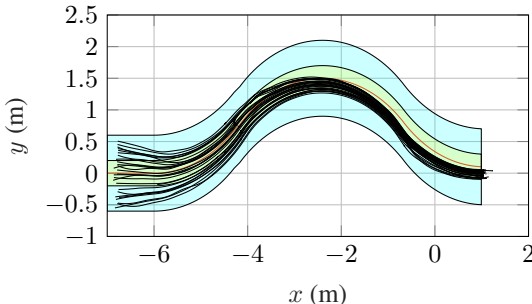

Figure 6: Experimental results where colors are as in Figure 4a. It is shown that if the platform's initial position is contained in the maximal ROA the platform will never exit the teal region.

controller as discussed in Chang et al. [17]). One MPC has a low cost of applying controls (MPCL) while the other one has a high cost of applying controls (MPCH). We compare our control method in straight- and curved-path following problems. The straight target path shown in Figure 7(a) is a straight line from 0 to 30 m, with an obstacle in the initial direction of the vehicle, just outside of the vehicle control funnel (as defined by the computed CLF in Figures 4 and 5). The initial position of the vehicle is on the target path, but oriented in yaw by the largest $\theta_e$ contained in the maximal ROA in Figure 4(a). This is an adversarial condition for non-verified controllers as we are unable to calculate the maximum path deviation. Rather, the amount of deviation from the target path will depend on design parameters such as control costs, MPC horizon and tuning constants. The curved target path is a circle with a radius of 4 m. The vehicle begins at $(0, 4)$ and completes one full lap of the path. Figure 7(b) shows the vehicle's deviation from the target path as a time series.

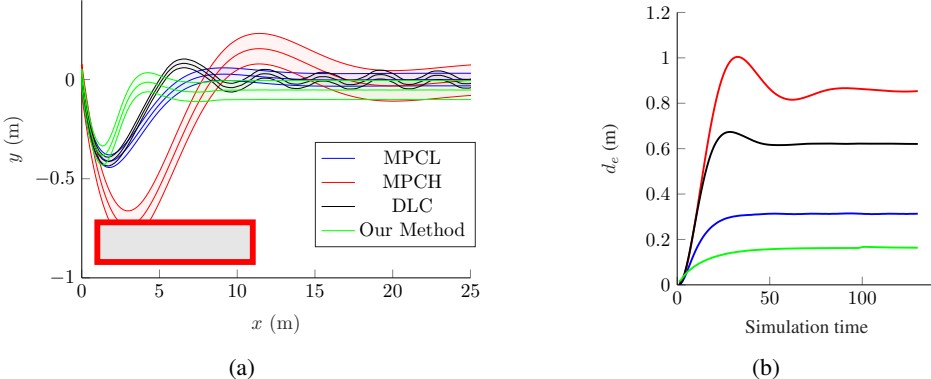

(a)                                          (b)

Figure 7: Comparison of control methods for (a) straight and (b) curved paths. (a) The red box represents an obstacle. (b) The $y$-axis represents the average deviation from the target path.

For each controller, we run 50 simulations with Gaussian noise ($\mu = 0$, $\sigma = 0.1$) added to the initial conditions $\theta_e$ and $d_e$. Figure 7 shows the test setup and simulated runs. Figure 7(a) shows the target paths with each colored area being the empirical average path (solid line in the middle) plus/minus one standard deviation.

As shown in Figure 7 and Table 1, our method is the only method that is both a verified method of control (guaranteed to avoid the obstacle outside of the control funnel) and able to stabilize to the target path (small steady state error). In particular, one will note that the other verified control method (DLC) oscillates around the target path, resulting in a large mean steady state error. Additionally our method provides the fastest convergence to the target, with a small maximum deviation, and is more than 2 orders of magnitude faster than the MPC implementation, making it a reasonable control choice even when verified control is not mandatory.

Table 1: Comparing performance and safety of verified and non-verified controllers over 50 simulations. "Average X" refers to the empirical average value of X over the 50 simulations, while "max X" and "mean X" refer to the maximum and mean value of X over a given simulation. The steady state error is calculated as the $L_1$ error upon the vehicle reaching steady state (after approximately 10 m as shown in Figure 7).

| | Average mean controller computation time (ms) | Safe paths | Average max $d_e$ (meters) | Maximum max $d_e$ (meters) | Average mean steady state error |
|---|---|---|---|---|---|
| **Straight Path (Figure 7(a))** | | | | | |
| Our Method | 0.75 | 50/50 | 0.39 | 0.56 | 0.0003 |
| DLC [17] | 0.08 | 50/50 | 0.42 | 0.49 | 0.0171 |
| MPCL [22] | 102 | 50/50 | 0.43 | 0.54 | 0.0046 |
| MPCH [22] | 99 | 12/50 | 0.74 | 0.98 | 0.0569 |
| **Curved Path (Figure 7(b))** | | | | | |
| Our Method | 1.25 | – | 0.16 | 0.29 | 0.16 |
| DLC [17] | 0.09 | – | 0.42 | 0.68 | 0.62 |
| MPCL [22] | 120 | – | 0.31 | 0.31 | 0.31 |
| MPCH [22] | 110 | – | 1.00 | 1.04 | 0.85 |

## 4 Limitations

The primary limitation of this work is the method to generate verified CLFs. Some desirable CLFs do not converge to a solution (such as CLFs with larger Lyapunov regions). The limiting factor is likely the verification phase, specifically the verification of Eq. (1). Rather than verifying $\exists\, u \in \mathcal{U}$ for which $\nabla V(x) \cdot f(x, u) < 0$, we write $u$ in terms of our LQR controller (a function of the system state) to provide a single $u$ for each state. Of course this greatly constrains the problem and means that the LQR controller constants must be trained simultaneously to the CLF parameters [17]. In theory SMT solvers should be able to directly check Eq. (1) as it is a first-order logic inequality, however in practice the control spaces are often too large. In future work one may be able to reduce the space by bounding $\mathcal{U}$ to some neighborhood of a sampled control. Additionally these verification techniques are conditioned on how well-represented a system is by the model $f(x, u)$. Namely, a model which is developed to represent certain behaviors of our system limit the extent of maneuvers that are verified and may be executed by the platform. This crucial limitation exists at the interface of *model validation*, wherein the adequacy of the model is evaluated. Possible next steps to address this limitation may be to consider online model synthesis and the extension of verification to these new models.

## 5 Conclusion

In this work we show the ability to generate CLFs using neural networks [17], extract executable controls to stabilize the system, and construct safe trajectories using a single CLF. Future work will seek improvements to CLF generation and verification. Additionally, CLFs can be used inside a receding horizon formulation in Algorithm 1 to provide potentially smoother control experience and faster stabilization to the target. Finally the feasibility of non-smooth maneuvering in this framework would provide another layer of robustness when operating in less ideal conditions.

**Acknowledgments**

This work was funded by NSF award #1932189 and the Belgian American Education Foundation (BAEF). All opinions expressed are those of the authors and not necessarily of the NSF or BAEF.

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
