# OpenReview forum: "Verified Path Following Using Neural Control Lyapunov Functions"
_robot-learning.org/CoRL/2022/Conference — CoRL 2022 Poster_

### Official Review · Reviewer_VUEm · 2022-06-27

**Originality:** Fair
**Technical Quality:** Fair
**Clarity Of Presentation:** Good
**Impact:** 3

**Recommendation:**

Weak Reject: I recommend rejecting the paper, but will not argue for my recommendation if the majority of other reviewers have a different opinion.

**Summary:**

This paper proposes a new way to do path following using neural network based control Lyapunov functions (CLF). For a given vehicle dynamic with straight-line reference paths, the authors construct the neural CLF to satisfy the Lyapunov conditions. Next they derive the curvature bounds to ensure the CLF controller stay valid. Hardware experiments are conducted to showcase the proposed approach's real-time performance.

**Issues:**

The only novelty I can sense from the paper is the way to derive a bound for the reference curvature so that one CLF can fit for different reference paths. This is good, but how much can we benefit from this? So my suggestion is the following:

(1) For the vehicle experiment, first compare this method to the classical controllers (PID, LQR, MPC...) and to show that this experiment is indeed a hard problem that previous methods cannot really solve it elegantly. And then compare this method with reinforcement learning (I suggest pick some candidates from DDPG, SAC, TRPO, PPO) to show that this is not just another gym environment that can be easily tackled by RL.

(2) Next, add different experiment settings. Compare with vanilla version of the CLF. Can your method converge under more initial conditions? What about the converging speed? How about the training time and model size? What happened if you try the curvature out of the bounded region?

(3) Finally, please try more experiments. One vehicle example in such low-dimension is not persuasive at all. You definitely want to consider more challenging cases (more complicated vehicle models, drones...)


**Quality Of The Limitations Section:**

Additional details required

**Reviewer Expertise:**

5: The reviewer is absolutely certain that the evaluation is correct and very familiar with the relevant literature

**Robotics Focus:**

Sufficient demonstration on hardware

**Strengths And Weaknesses:**

Strengths: theoretical guarantees for the CLF converging behavior and tolerant reference curvature bound ; hardware implementation to showcase the proposed controller can work in real-time.

Weaknesses: this work is incremental since there are already so many papers in using NN to construct CLF, and no baselines used in the experiments to demonstrate what is the benefit of this new NN CLF method. And they only use this CLF controller in a relative low-dimension experiment. The verification process can only converge within a small RoA.

**Summary Of Recommendation:**

My decision is weak reject. In the past few years we have witnessed so many papers in using neural networks to construct the Control Lyapunov Function, and I don't think this paper brings enough contribution to this field. The experiment is so simple and there aren't enough baselines or different setups to demonstrate the benefit of the proposed approach.

---

> ### Author Response · Authors · 2022-08-20
> **Review Response with Addition of Side by Side Comparison of Alternative Techniques**
>
> Thanks for the feedback.
>
> Regarding the novelty of the paper, the main contribution is to provide a method for deriving a CLF for curved path following from one for straight path following and show that this provides a flexible and efficient approach for complex path following tasks (see also the first paragraph of the response to the Area Chair for details). This work is, to the best of our knowledge, the first one to propose such an approach.
>
> We have added experiments to compare our approach with state-of-the-art “verified” and “non-verified” approaches (see section 3.3 of the attached pdf and the second paragraph of the response to the Area Chair for details). These experiments show that our approach is favorable in terms of safety and computation speed, as well as the ability to stabilize to a target path, when compared to these approaches. In the added experiments, we have also considered configurations of the platform outside of the region of attraction and observed that the controlled platform still converges to the path, thereby demonstrating the practical robustness of the approach with respect to initial conditions and perturbations.
>
> It is true that the model considered in the paper is relatively simple and that the derivation of the curved path following CLF is rather model-specific. Nevertheless, this provides a proof-of-concept that mixed learning (straight-path CLF generation) and analytical (promotion of the CLF for curved path following) can be used as a flexible and efficient approach for complex path following tasks, and we demonstrate this with simulation and concrete experiments. We believe that this approach can be leveraged for more complex vehicle models involving (direct or indirect) steering control.

---

### Official Review · Reviewer_mRSR · 2022-07-31

**Originality:** Good
**Technical Quality:** Good
**Clarity Of Presentation:** Very Good
**Impact:** 4

**Recommendation:**

Weak Reject: I recommend rejecting the paper, but will not argue for my recommendation if the majority of other reviewers have a different opinion.

**Summary:**

This paper proposes a framework to search for the best control for path tracking problems using neural control Lyapunov functions. The authors show the proposed method can ensure a bounded deviation of a target path and demonstrate the effectiveness of their algorithm on a real robot.

**Issues:**

Does the process encounter issues of scalability?

**Quality Of The Limitations Section:**

Additional details required

**Reviewer Expertise:**

5: The reviewer is absolutely certain that the evaluation is correct and very familiar with the relevant literature

**Robotics Focus:**

Sufficient demonstration on hardware

**Strengths And Weaknesses:**

Strengths:

• This paper is well-written and easy to read.

• The framework provides a general and simple way to extract a stabilizing feedback controller.

Weaknesses:

• In the experiment section, there is lacking comparison between the proposed method and existing path tracking methods. It would be also interesting to see the performance comparison in controllers extracted by polynomial CLFs and neural CLFs.



**Summary Of Recommendation:**

Overall, this paper contributes to the current research field of search for a reliable controller with a theoretical guarantee and shows the process is practical in real robotic systems. However, it’s unclear to see how effective the proposed method is because of an absence of comparison.

---

> ### Author Response · Authors · 2022-08-20
> **Review Response with Addition of Side by Side Comparison of Alternative Techniques**
>
> Thanks for the feedback.
>
> We have added experiments to compare our approach with state-of-the-art “verified” and “non-verified” approaches (see section 3.3 of the attached pdf and the second paragraph of the response to the Area Chair for details). These experiments show that our approach is favorable in terms of safety and computation speed, as well as the ability to stabilize to a target path, when compared to these approaches.

---

> > ### Comment · Reviewer_mRSR · 2022-08-25
> > **Thanks for the detailed explanation and comparison**
> >
> > Thanks for the detailed explanation and comparison. Considering the authors stated in the response, the main novelty is to derive a CLF for curved path following from one for straight path following. It would be helpful to evaluate the contribution if the authors demonstrate the curved path following in the additional comparison. Thanks.

---

> > > ### Author Response · Authors · 2022-08-27
> > > **Curved Path Following Additional Comparison**
> > >
> > > Thank you, agreed that an additional comparison of the methods along a curved path would be beneficial for further evaluating the control methods. We will add this comparison in addition to the straight line comparison to the camera ready version if accepted. In a quick test of these control methods following a curved path, the metrics in table 1 are very similar, the only major difference is that the verified learned control (DLC) no longer oscillates around the target path, rather it drifts away from target path.

---

### Official Review · Reviewer_wkL6 · 2022-07-31

**Originality:** Poor
**Technical Quality:** Very Good
**Clarity Of Presentation:** Very Good
**Impact:** 2

**Recommendation:**

Weak Reject: I recommend rejecting the paper, but will not argue for my recommendation if the majority of other reviewers have a different opinion.

**Summary:**

This paper applies existing techniques on learning Control Lyapunov Function (CLF) to the application of path following of ground vehicles. The main contribution of the paper is the ability to safely use a CLF designed for low curvature paths to paths with higher curvatures without losing the correctness guarantees of the CLF. The paper provides sufficient experimental evidence to support the proposed approach.

**Issues:**

- The main technical problem the paper aims to solve is unclear. Does the paper aims to answer the question of "how to obtain a CLF for ground vehicles?". But such a problem can be analyzed manually, and a learning-based approach is not justified. Or does the paper aims to answer the question of "Is neural-based CLF practical and can it be used for real-world example?" for which the use of ground vehicles is just an example to show these limitations? However, to answer this question one would expect the use of neural-CLF on a diverse set of problems to understand its limitations. Or does the paper aims to answer "how to use CLFs designed for paths with low curvatures to obtain guarantees on paths with higher curvature?" and the neural-based CLF happens to be just an example of a CLF?

- It is unclear how the proposed framework handles the mismatch between the dynamical model (equation 4) and the actual robotic platform. The numerical results section does not describe the effect of this mismatch and applies the CLF designed for the nominal model directly to the platform.

- How does the proposed approach compare against other non-learning approaches like using MPC with CLF constraints?

**Quality Of The Limitations Section:**

Limitations are addressed clearly

**Reviewer Expertise:**

4: The reviewer is confident but not absolutely certain that the evaluation is correct

**Robotics Focus:**

Sufficient demonstration on hardware

**Strengths And Weaknesses:**

Strengths:
- The paper applies the proposed framework to a real-world example with interesting experimental results.

Weaknesses:
- The paper's contribution is limited to applying existing algorithms (i.e., neural-based CLF learning and verification) with minor modifications.

- Control Lyapunov functions for ground vehicles have been extensively studied, and several manually designed CLFs have been discussed in the literature. The paper lacks a clear justification of the motivation behind using a neural-based CLF for this setup.

- The paper lacks a comparison against other techniques to understand the advantages and disadvantages of using the proposed framework.



**Summary Of Recommendation:**

While the paper provides interesting experimental results on a real platform, the novelty of the paper is limited to applying existing algorithms of neural-based CLF with minor modifications.

---

> ### Author Response · Authors · 2022-08-20
> **Review Response with Addition of Side by Side Comparison of Alternative Techniques**
>
> Thanks for the feedback. Addressing the weaknesses:
>
> Regarding the contribution of the paper, please see the first paragraph of the response to the Area Chair.
>
> Regarding the use of NN, as we explain in the first paragraph of the response to the Area Chair, we picked this approach as one possible approach to compute a CLF for straight path following, but in further work we will consider other approaches, namely to optimize some characteristics of the computed CLF, like the region of attraction, the speed of convergence and/or the bound on the allowed curvature.
>
> We have added experiments to compare our approach with state-of-the-art approaches, including “verified” and “non-verified” approaches (see section 3.3 of the attached pdf); please see the second paragraph of the response to the Area Chair for details.
>
> Thank you again for the feedback, to address the Issues:
>
> The authors acknowledge this gap, however the experimental validation of this technique being embodied on a ground vehicle does not limit its applicability. Many techniques developed with wide application domains only demonstrate on a single experimental platform. Most importantly, the authors disagree with the assertion that applicability limited to ground-based vehicles is unjustified. See for instance Williams et al.*; feedforward dynamics can be very rapidly computed using neural networks, which offers a distinct advantage over other modeling techniques. If accepted, we will highlight more clearly the contribution of the paper, which is to obtain CLFs for curved path following from CLF for straight/low-curvature path following (see the first paragraph of the response to the Area Chair for details), and we will add comparisons with other approaches (see the second paragraph of the response to the Area Chair for details).
> * G. Williams et al., "Information theoretic MPC for model-based reinforcement learning," 2017 IEEE International Conference on Robotics and Automation (ICRA), 2017, pp. 1714-1721, doi: 10.1109/ICRA.2017.7989202.
>
> It is true that we do not quantify the effect of a mismatch between the model and the platform. In Section 4 (paragraph 2), we mention this limitation and propose approaches for handling it to investigate in future work. On the other hand, the experiments show that the CLF derived from the model is able to control the actual platform, thereby demonstrating the robustness of the approach to some extent. In the added experiments (see the second paragraph of the response to the Area Chair for details), we have also considered configurations of the platform outside of the region of attraction and observed that the controlled platform still converges to the path, thereby demonstrating the practical robustness of the approach with respect to initial conditions and perturbations.
>
> The added experiments (section 3.3 of the attached pdf; see also the second paragraph of the response to the Area Chair) show that our approach is favorable in terms of safety and computation speed, as well as the ability to stabilize to a target path, when compared to state-of-the-art “verified” and “non-verified” approaches.

---

### Official Review · Reviewer_U6aE · 2022-08-01

**Originality:** Good
**Technical Quality:** Good
**Clarity Of Presentation:** Good
**Impact:** 4

**Recommendation:**

Weak Accept: I recommend accepting the paper, but will not argue for my recommendation if the majority of other reviewers have a different opinion.

**Summary:**

This paper considers the problem of tracking a curve path for a vehicle model. The authors apply the neural Lyapunov function approach to a straight line tracking problem. Given the learned Lyapunov function, the authors compute the largest curvature that the Lyapunov function can tolerate. The idea of this paper is straightforward, but the authors evaluated it on hardware.

**Issues:**

1. The definition of u_{req} in Line 172 is confusing. Maybe move Eq. (5) here?
2. Caption of figure 2 is too close to the text. Maybe remove the blank space on the top and bottom of the figure?
3. L124: verified -> be verified.
4. Explain the function in Algorithm 1.

**Quality Of The Limitations Section:**

Limitations are addressed clearly

**Reviewer Expertise:**

3: The reviewer is fairly confident that the evaluation is correct

**Robotics Focus:**

Sufficient demonstration on hardware

**Strengths And Weaknesses:**

Strengths:
1. Experiments on hardware.

Weakness:
1. Theoretical contribution is minor.
2. Some presentation issues.

**Summary Of Recommendation:**

Although the idea is simple, the experiments on hardware look promising.

---

> ### Author Response · Authors · 2022-08-20
> **Review Response with Further Description of Theoretical Contribution**
>
> > Theoretical contribution is minor.
>
> Regarding the theoretical contribution of the paper, please see the first paragraph of the response to the Area Chair.
>
> > Some presentation issues.
>
> Thanks for this feedback. If accepted, we will significantly improve the presentation of the paper for the final version. In particular, we will highlight more clearly the contribution of the paper (as explained in point 1) and present comparisons with other approaches (see also the second paragraph of the response to the Area Chair).

---

### Meta-Review · Area_Chair_R3Z5 · 2022-08-13

**Recommendation:** Accept (Poster)
**Confidence:** 4

**Metareview:**

The paper focuses on path following of a ground vehicle with the use of NN-based CLF. It presents appealing demonstrations in hardware, however it lacks empirical comparisons to prior works. The paper is generally well-written and technically clear, but the theoretical novelty is limited as the approach relies on minor modifications to existing methods. Clearer justification of the significance is needed.

Main pros:
- Promising hardware experiments
- The paper is well-written

Main cons:
- Lack of comparison to baselines in the experiments
- Limited theoretical novelty

Post-rebuttal update: The revised version of the paper has offered comparisons to baselines. The authors also provided further explanations regarding the theoretical novelty. It is important that the final version of the paper will show also experiments with curved path following as the authors discussed with reviewer mRSR.

**Best Paper Nomination:**

No

---

> ### Author Response · Authors · 2022-08-20
> **Comment Response and Updated Document Upload**
>
> The authors thank the area chair for their feedback.
>
> To reiterate what we stated in the manuscript, the main novelty of our approach is to derive a CLF for curved path following from one for straight path following, and furthermore do this using data-driven techniques. Proving this is not a trivial matter, and is crucial for remaining within the verified control framework. We demonstrate the success of this method beyond related previous approaches in our Results. This work is, to the best of our knowledge, the first to employ such an approach. Finding a CLF for a general path is in general difficult; furthermore, most available techniques present flexibility issues as the CLF needs either to be recomputed at every update of the path, or built from a (by definition limited) library of CLF primitives [18]. In contrast, our approach allows us to follow curved paths, possibly updated on-the-fly, as long as the curvature is bounded by some a priori computed constant. This allows for great flexibility in the path planning, e.g., for emergency obstacle avoidance tasks. Finally, by restricting the computation to that of a single CLF (the one for straight path following), we can consider several costs and objectives in the computation of the CLF, like maximizing the region of attraction, the speed of convergence and/or the bound on the allowed curvature (doing the same with a lot of CLFs, or at each update of the path could be very expensive in term of computation). In the current version of this work, we have restricted ourselves to computing a valid CLF (using NN) and deducing the values of the above characteristics a posteriori, but in further work, we will incorporate the optimization of these characteristics in the computation of the CLF.
>
> We have added several experiments to compare our approaches with state-of-the-art approaches (see Section 3.3 of the attached pdf). In particular, we have compared with “non-verified” approaches (MPC [26]) and “verified” approaches (verified Neural Network controller proposed in [17]). The experiments show that our approach is favorable in terms of safety (some “non-verified” approaches lead to collisions) and computation speed, as well as the ability to stabilize to a target path, when compared to these state-of-the-art approaches.